# Urinary Tract Infections among Febrile Infants in Qatar: Extended-Spectrum-Beta-Lactamase (ESBL)-Producing Versus Non-ESBL Organisms

**DOI:** 10.3390/antibiotics13060547

**Published:** 2024-06-12

**Authors:** Mohammad Qusad, Ihsan Elhalabi, Samer Ali, Khaled Siddiq, Lujain Loay, Abdallah Aloteiby, Ghada Al Ansari, Bassem Moustafa, Tawa Olukade, Mohammed Al Amri, Ashraf Soliman, Ahmed Khalil

**Affiliations:** 1Section of Academic General Pediatrics, Department of Pediatrics, Hamad General Hospital, Doha 3050, Qatar; mqusad@hamad.qa (M.Q.); ielhalabi@sidra.org (I.E.); sali8@sidra.org (S.A.); ksiddiq@hamad.qa (K.S.); lloay@sidra.org (L.L.); aaloteiby@sidra.org (A.A.); bmoustafa@hamad.qa (B.M.); 2Department of Clinical Chemistry Laboratory, Hamad General Hospital, Doha 3050, Qatar; galanssari@hamad.qa; 3Department of Pediatrics, Hamad General Hospital, Doha 3050, Qatar; tolukade@hamad.qa (T.O.); malamri@hamad.qa (M.A.A.); atsoliman56@gmail.com (A.S.); 4Section of Pediatric Clinical Pharmacy, Pharmacy Department, Hamad General Hospital, Doha 3050, Qatar

**Keywords:** prevalence, urinary tract infection, febrile infants, ESBLs

## Abstract

Background: The escalating prevalence of ESBL-producing Enterobacteriaceae in Qatar’s pediatric population, especially in community-onset febrile urinary tract infections (FUTIs), necessitates a comprehensive investigation into this concerning trend. Results: Over the course of one year, a total of 459 infants were diagnosed and subsequently treated for UTIs. Cases primarily occurred in infants aged over 60 days, predominantly non-Qatari females born from term pregnancies. Notably, *E. coli* and *K. pneumoniae* were the most frequently identified organisms, accounting for 79.7% and 9.8% in the ESBL group and 57.2% and 18.7% in the non-ESBL group, respectively. Interestingly, hydronephrosis emerged as the most prevalent urological anomaly detected in both ESBL (*n* = 10) and other organism (*n* = 19) groups. Methods: In this retrospective cohort study conducted in Qatar, we meticulously evaluated the prevalence of pediatric FUTIs. Our study focused on febrile infants aged less than 1 year, excluding those with urine samples not obtained through a catheter. Conclusions: *E. coli* and *K. pneumoniae* prevailed as the predominant causative agents in febrile children in Qatar, with hydronephrosis being identified as the most common urological anomaly. Moreover, our findings suggested that gentamicin served as a viable non-carbapenem option for hospitalized ESBL cases, while oral nitrofurantoin showed considerable promise for uncomplicated ESBL UTIs.

## 1. Introduction

Urinary tract infections (UTIs) are prevalent among children [1,2]. Up to 11% of children have experienced a UTI by the age of 16, with girls having a greater infection incidence than boys [3]. In pediatric clinical practice, the most prevalent confirmed bacterial infections are febrile urinary tract infections (FUTIs). FUTIs can cause significant morbidity and long-term consequences, such as renal scarring, hypertension and chronic renal failure [4,5]. Early detection and proper treatment decrease the probability of renal scarring and its related complications [6].

Extended-spectrum beta-lactamases (ESBLs) are a class of plasmid-encoding enzymes generated mostly by enterobacteria. The advent of extended-spectrum-β-lactamase-producing Enterobacteriaceae (ESBL-E) as a cause of FUTIs poses a severe concern to public health, with limited treatment options [7,8]. A few years ago, ESBL-E was primarily isolated in hospitals and other healthcare institutions.

The prevalence of ESBLs in urine among Qatar’s pediatric population has gradually increased (HMC yearly antibiogram) [9]. However, such organisms have disseminated throughout the population, and the prevalence of community-onset FUTIs caused by ESBL-E isolates has grown globally [10,11]. Not only ESBLs, but also uropathogenic *Escherichia coli* (*E. coli*) is responsible for 80–90% of pediatric UTIs [12], and the majority of FUTIs [13,14].

In Gulf Cooperation Council (GCC) countries (including Qatar), isolates producing ESBLs show resistance not only to cephalosporins and monobactams, but also to other antibiotic classes like aminoglycosides and fluoroquinolones [15]. Also, *E. coli* urine isolates from children in the GCC area (including Qatar) are highly resistant to popular antibiotics used to treat UTIs, such as ampicillin and trimethoprim/sulfamethoxazole [15].

The prevalence of urinary tract infections (UTIs) among febrile children in Qatar has not been comprehensively addressed in the existing literature. Thus, this study aims to provide a comprehensive assessment of the current prevalence of pediatric febrile UTIs (with a particular focus on extended-spectrum-beta-lactamase-producing organisms) in Qatar. It seeks to compare clinical histories, laboratory parameters and ultrasound findings related to urological abnormalities, as well as the types of antibiotics used and their routes of administration between ESBL and non-ESBL organisms.

## 2. Results

For one year, 459 infants were diagnosed and treated for UTIs. The study identified a prevalence of 16.1 ESBL-producing UTI cases per 1000 pediatric admissions among infants with suspected urinary tract infections. Most of these infants (*n* = 403, 87.8%) were aged over 60 days at the time of presentation, non-Qatari (*n* = 379, 82.6%) and female (*n* = 298, 65%). Most infants (87.5%) were born from term pregnancies and 31.7% of male infants were circumcised, as detailed in (Table 1).

The most frequently isolated uropathogens were *Escherichia coli* and *Klebsiella pneumoniae*. *Escherichia coli* predominated as the primary ESBL uropathogen (79.7%). No significant differences were observed in the demographic variables between the ESBL and non-ESBL organism groups (Table 2).

Table 3 presents the findings about significant clinical histories, laboratory parameters and imaging abnormalities detected in US or VCUG among patients infected with ESBL-producing organisms compared to those infected with non-ESBL pathogens. Several of these findings were clinically relevant. Notably, the ESBL group demonstrated a higher prevalence of urine WBCs compared to non-ESBL organisms (80.8% vs. 64.9%, respectively; *p* < 0.001), as per clinical guidelines. This pattern was also observed for the urine analysis of leukocytes (54.2% vs. 42.5%, *p* = 0.023) and nitrites (38.9% vs. 24.2%, *p* = 0.002), respectively. While there was a noticeable difference in creatinine levels between the two groups, it did not have much clinical impact, as it remained within normal ranges for their respective age groups. Moreover, although not statistically significant, it is worth mentioning that the ESBL group tended to have higher median values for inflammatory markers compared to the group infected with non-ESBL organisms. These markers included the white blood cell count (13.3 vs. 12), absolute neutrophil count (5.8 vs. 4.9), C-reactive protein (29 vs. 17.4) and platelet count (407 vs. 373). Additionally, there were no significant differences observed in other measured factors, such as admission to the neonatal intensive care unit (NICU), previous hospitalizations or surgeries, known urinary system abnormalities, abnormalities detected using VCUG or US, circumcision status among male infants or previous positive urine cultures.

The US evaluation of the kidneys and urinary tract revealed abnormalities in only 35 cases (11 within the ESBL group and 24 among those infected with non-ESBL organisms). Hydronephrosis emerged as the most common pathology, followed by dysplastic kidney, horseshoe kidney, ectopic kidney and duplex collecting system, as outlined in (Table 4).

Table 5 outlines the utilization of antibiotics (expressed in percentages) among infants diagnosed with ESBL infections compared to those with infections from non-ESBL organisms, including their administration routes. Overall, a higher proportion of infants in the ESBL group (56.5%) received intravenous (IV) antibiotics compared to those infected with non-ESBL organisms (34.4%). In the high-risk, complicated ESBL UTI group, gentamicin (36.4%) emerged as the primary antibiotic choice, followed by nitrofurantoin (11.4%) for noncomplicated cases managed as outpatients in the emergency department. Empirical treatment with ceftriaxone (9.1%) and cefixime (22.7%) was initiated depending on the severity of the case and hospitalization status, with subsequent adjustments based on sensitivity results to gentamicin and nitrofurantoin, respectively. In contrast, for the non-ESBL organism group, cefixime (45.1%) was the predominant oral antibiotic prescribed for outpatient cases, then amoxicillin/clavulanate (15.4%), while ceftriaxone (21.9%) was the most used IV antibiotic for hospitalized inpatient cases.

## 3. Discussion

Through this retrospective cohort study, we highlighted the prevalence of pediatric FUTIs and compared clinical histories, laboratory parameters, US urological abnormalities and the antibiotics used and their routes of administration between ESBL and non-ESBL organisms among infants who were admitted to the Hamad General Hospital (pediatric emergency center and inpatient unit). The study identified a prevalence of 16.1 ESBL-producing UTI cases per 1000 pediatric admissions among infants with suspected urinary tract infections (UTIs). Hanna-Wakim et al. evaluated hospitalized children over ten years and reported a positivity ESBL rate of 15.5%, which closely aligned with our findings [13]. Additionally, a recent systematic review and meta-analysis revealed a pooled prevalence of pediatric ESBL UTIs of 5% in Eastern Mediterranean studies, contrasting with an overall prevalence of 14% in all other countries [16]. This indicated regional variations in ESBL prevalence. Notably, some regions reported significantly higher rates; for instance, a cohort study from Turkey and Jordan documented prevalence rates as high as 46% and 49.3%, respectively [17,18]. These higher rates could be attributed to differences in antibiotic usage patterns, healthcare practices and the effectiveness of infection control measures. Our study’s prevalence fell within the range reported by other studies, highlighting the ongoing global challenge of managing ESBL-producing bacterial infections in pediatric populations. These findings underscored the importance of continuous surveillance, effective antibiotic stewardship and robust infection control measures to mitigate the impact of these resistant pathogens.

With regard to the demographic characteristics, most of the infants were female in both the ESBL group (65.4%) and non-ESBL organism group (64.7%). Similarly, Awean GZA et al.’s study in Qatar showed that most of the infected patients were female in ESBL bacteria (78%) and non-ESBL bacteria (76.8%) [19], while Han et al.’s study showed that most of the infected patients were male in the ESBL group (52.4%) and non-ESBL group (60%) [20].

The nationality distribution, which was similar to the overall distribution, showed that most of infants were non-Qatari in the ESBL group (85.7%) and in the non-ESBL organism group (81.3%). Likewise, Awean GZA et al.’s study showed that most of the infected patients were non-Qatari in the ESBL group (85.3%) and non-ESBL group (75.3%) [19].

In terms of the infants’ age distribution, the majority of infants in both the ESBL group (91%) and the non-ESBL organism group (86.5%) were over 2 months old, aligning with established trends. Notably, the study by Park SY and Kim JH revealed that the mean age of infants in the ESBL group (4.31 months old) was younger than those in the non-ESBL group (6.26 months old). Conversely, Kim YH et al. found that the mean age of infants in the ESBL group (4.2 months old) was older than in the non-ESBL group (3.1 months old) [21,22]. Furthermore, Awean GZA et al. observed that the majority of infected infants were under 1 year old in both the ESBL group (45.6%) and the non-ESBL group (39.2%) [19]. Additionally, a trend regarding a higher incidence of ESBL cases in children less than a year old was noted, which was consistent with findings from two studies conducted in Turkey and Jordan [17,18].

In examining the prevalence of urinary organisms, *E. coli* and *Klebsiella pneumoniae* (*K. pneumoniae*) emerged as the predominant strains encountered within the ESBL group, accounting for 79.7% and 9.8% of cases, respectively. Meanwhile, within the group of other organisms, *E. coli* and *K. pneumoniae* were also prevalent, though to a lesser extent, constituting 57.2% and 18.7% of cases, respectively. When exploring existing research, findings from the study conducted by Park SY and Kim JH underscored *E. coli*’s predominance as the most frequently encountered organism in both ESBL and non-ESBL groups, accounting for 100% and 89% of cases, respectively. Following *E. coli*, *K. pneumoniae* and *Enterobacter* spp. were noted in the non-ESBL group, comprising 4% and 3% of cases, respectively. Similarly, Han et al. corroborated these trends, indicating *E. coli* and *K. pneumoniae* as the leading organisms in both ESBL and non-ESBL groups, with proportions of 42.9% and 50% in the former and 50% each in the latter. Aligning with these observations, the study conducted by Awean GZA et al. in Qatar revealed a similar pattern, with *E. coli* and *Klebsiella* species predominating in both ESBL and non-ESBL groups, albeit with varying proportions. Specifically, *E. coli* and *Klebsiella* species were identified in 32.4% and 32% of cases in the ESBL group and in 67.6% and 68% of cases in the non-ESBL group, respectively [19,20,21].

Regarding laboratory parameters, the WBC median was higher within the ESBL group (13.3 × 10^3^/mm^3^) compared to the non-ESBL organism group (12 × 10^3^/mm^3^). Notably, the analysis revealed that the differences in the WBC counts between the groups were not significant, with values being similar. This was consistent with findings from Park SY and Kim JH’s study, which showed comparable WBC means between the ESBL group (13.4 × 10^3^/μL) and the non-ESBL group (13.7 × 10^3^/μL). Similarly, Kim YH et al.’s study reported similar WBC means in the ESBL group (14.4 × 10^3^/mm^3^) and the non-ESBL group (15.5 × 10^3^/mm^3^), and N.C. Fan et al.’s study also found comparable WBC means between the ESBL group (12,517.1/μL) and the non-ESBL group (13,304.4/μL) [21,22,23].

The median platelet count among infants in the ESBL group was (407 × 10^3^/μL) compared to (373 × 10^3^/μL) in the non-ESBL organism group. However, this disparity in platelet counts between the groups did not reach statistical significance. These observations aligned with the conclusions drawn in Park SY and Kim JH’s study, which highlighted the comparability of platelet means between the ESBL group (442 × 10^3^/μL) and the non-ESBL group (428 × 10^3^/μL). Similarly, N.C. Fan et al.’s study also corroborated these findings, reporting analogous platelet means between the ESBL group (348 × 10^3^/μL) and the non-ESBL group (342 × 10^3^/μL) [21,23].

For creatinine, we observed a similarity in levels between the two groups. Among infants in the ESBL group, the median creatinine level (20 umol/L) closely approximated that of the other organism group (22 umol/L). These findings were in line with the results reported by Park SY and Kim JH, where the mean creatinine levels exhibited comparability between the ESBL group (20.3 umol/L) and the non-ESBL group (19.4 umol/L). However, it is noteworthy to mention that N.C. Fan et al.’s study recorded higher creatinine levels in both groups, with the ESBL group (35 umol/L) and the non-ESBL group (53 umol/L) demonstrating elevated values compared to those observed in our study [21,23].

We observed a higher median level of C-reactive protein (CRP) among infants in the ESBL group (29 mg/L) compared to the group infected with non-ESBL organisms (17.4 mg/L). These results mirrored the findings of previous research conducted in Park SY and Kim JH’s study, which similarly reported elevated CRP levels in the ESBL group (45.5 mg/L) compared to non-ESBL group (43.5 mg/L). However, it is noteworthy that the observed differences did not reach statistical significance in either our study or theirs [21].

The number of previous hospitalizations significantly differed among infants in the ESBL group (*n* = 12) compared to those infected with non-ESBL organisms (*n* = 26), although this difference was not statistically significant. This observation was consistent with findings from Park SY and Kim JH’s study, which similarly reported a lower number of previous hospitalizations in the ESBL group (*n* = 10) compared to the non-ESBL group (*n* = 39). It is important to note, however, that in their investigation, this variation reached statistical significance with a *p*-value of 0.02 [21].

The presence of urological anomalies identified via voiding cystourethrogram (VCUG) was less frequent among infants in the ESBL group (*n* = 3) compared to those infected with non-ESBL organisms (*n* = 6). Similarly, Awean GZA et al.’s study demonstrated a lower occurrence of urological anomalies via VCUG in the ESBL group (*n* = 6) compared to the group infected with non-ESBL organisms (*n* = 8) [19]. Moreover, the presence of urological anomalies identified via ultrasonography (US) was less frequent among infants in the ESBL group (*n* = 11) compared to those in the non-ESBL organism group (*n* = 24). Awean GZA et al.’s study also supported this observation, showing a lower presence of urological anomalies via US in the ESBL group (*n* = 11) compared to the group infected with non-ESBL organisms (*n* = 18) [19]. When examining specific urological anomalies identified via US, hydronephrosis emerged as the most prevalent anomaly among infants in both the ESBL group (*n* = 10) and the non-ESBL organism group (*n* = 19). This finding aligned with the results of Park SY and Kim JH’s study, where hydronephrosis was similarly prevalent among both the ESBL group (*n* = 7) and the non-ESBL organism group (*n* = 36) [21].

Based on our analysis of antibiotic usage, we found that both oral and intravenous routes were commonly employed in treating infants. In the ESBL group, antibiotics were predominantly administered intravenously (56.9%), whereas in the non-ESBL organism group, they were administered via both oral (65.6%) and intravenous (34.4%) routes. Inpatient treatment protocols for infants aged younger than 2 months typically comprised intravenous third-generation cephalosporins, notably ceftriaxone and cefotaxime, targeting non-ESBL organisms, consistent with the findings reported by Zorc et al. and in other studies [14,24,25]. Aminoglycosides, including gentamycin and amikacin, demonstrated complete sensitivity throughout the study period. Notably, gentamycin emerged as the most frequently administered non-carbapenem antimicrobial agent for ESBL cases, followed by amikacin, corroborating earlier studies by Jo et al. and Joo and Shin [26,27]. In older infants aged over 2 months, oral cefixime and amoxicillin/clavulanate emerged as the most prescribed antibiotic for non-resistant organisms, in line with our observation. For noncomplicated ESBL cases, oral treatments like nitrofurantoin and sulfamethoxazole/trimethoprim were utilized, mirroring the findings of studies conducted by Raman G. et al., Robinson JL et al. and Wang ME et al. [28,29,30].

While this study provided valuable insights, it is essential to acknowledge its limitations. The data reviewed were confined to a single medical center and spanned only one year, potentially limiting their generalizability to the broader national landscape of microbial etiology and antibiotic resistance in ESBL UTIs. Nonetheless, this study holds significance, as it represents a pioneering effort in characterizing long-term surveillance data for UTIs among all pediatric patients in a tertiary referral hospital in Qatar. Notably, our center stands as the largest pediatric facility in Qatar, further underscoring the importance of this study in addressing a critical gap in understanding and managing UTIs in this population.

## 4. Methods

### 4.1. Study Design, Setting and Participants

This retrospective cohort study was based on routinely collected hospital data of febrile infants over a 12-month period from January to December 2019. The population consisted of infants aged less than one year. The investigation was primarily conducted at Hamad General Hospital (inpatient pediatric unit) and the El-Sad center, the largest emergency center in the state of Qatar, a cosmopolitan hub where a diversity of patients from different countries reside. In 2019, a total of 8237 pediatric patients were admitted to the hospital. The study included 459 of these pediatric patients.

The identification and susceptibility profiling of all bacteria from positive urine cultures were determined by using an automated susceptibility system VITEK 2 (Bio Merieux, Marcy l Etoile, France), according to the Clinical and Laboratory Standard Institute guidelines (CLSI).

The inpatient treatment at the UTI ward for infants aged 0–1 month with a febrile UTI or pyelonephritis recommended a treatment duration of 10–14 days. For infants older than 1 month with bacteremia due to urosepsis, there is no evidence suggesting that a prolonged course of parenteral antibiotics reduces the chance of a relapse. These infants can transition to oral antibiotics once they become afebrile and a repeat blood culture remains negative for 48 h. For uncomplicated UTIs in this age group, the treatment course is typically 5–7 days. However, for complicated UTIs, pyelonephritis, or in cases where the child is 2 months or younger, a 10–14-day treatment course is necessary. Complicated UTIs include conditions such as UTIs in the presence of renal calculi, in immunocompromised hosts or those presenting with severe illness like septic shock [31,32].

### 4.2. Inclusion and Exclusion Criteria

The study included patients who met the following inclusion criteria: febrile infants exhibiting symptoms suggestive of a urinary tract infection (UTI) and undergoing urine sample collection for microscopy and culture before commencing antibiotic treatment; positive findings of pyuria and bacteriuria on urine microscopy coupled with leukocyte esterase and nitrites positive on a urine dipstick analysis; additionally, culture results meeting the defined thresholds for a UTI, such as >50,000 colony-forming units per milliliter (cfu/mL) in catheterized samples or any growth in suprapubic aspiration [31,32].

The exclusion criteria included infants previously treated with antibiotics before urine sample collection, potentially leading to false negative results. Additionally, cases lacking pyuria and bacteriuria in urine microscopy and samples collected via a bag instead of a catheter were excluded due to potential contamination and unreliable results.

### 4.3. Data Collection and Study Variables

We retrieved data relating to patients’ demographics, past medical history, treatment given and laboratory and radiology data related to this study objective from electronic medical records. We extracted demographic variables of age and grouped participants as less than or equal to 60 days old or greater than 60 days of age at presentation, nationality as Qatari and non-Qatari and gender as male or female. We retrieved past medical histories of gestational age, neonatal intensive care unit (NICU) admission, circumcision status for males and history of previous UTIs or hospital admissions including the history of coexisting urogenital abnormalities. The laboratory parameters retrieved included the hemoglobin level, total white blood cell count (WBC), absolute neutrophil count (ANC), platelet count, C-reactive protein (CRP), sodium, blood urea nitrogen (BUN), creatinine (Cr) levels, urine WBC, nitrite and leukocytes.

Results for the urine extracted were grouped as ESBL-producing uropathogens or non-ESBL organisms. The non-ESBL group comprised both Gram-negative and other Gram-positive organisms detected in the urine culture for our pediatric cases. Finally, we extracted results for voiding cystourethrogram (VCUG) and US, the antibiotic used and the route utilized. Most variables were studied as dichotomized variables. The comparison groups were the ESBL versus non-ESBL organisms group from urine cultures.

### 4.4. Ethical Approval

The study was approved by the institutional review boards of the Hamad Medical Corporation (MRC-01-22-753). The requirement for consent was waived because of the retrospective nature of the study.

### 4.5. Statistical Analysis

We summarized the distribution of variables using numbers and percentages, and median and interquartile ranges as appropriate. To compare the ESBL with non-ESBL organisms, we used chi-square tests or the Mann–Whitney U test. A statistical analysis was performed using IBM SPSS 28 statistical software with a statistical significance set at *p* < 0.05.

## 5. Conclusions

*Escherichia coli* (*E. coli*) and *Klebsiella pneumoniae* (*K. pneumoniae*) emerged as the predominant organisms encountered in extended-spectrum beta-lactamase (ESBL) and other bacterial groups among febrile children in Qatar. Notably, hydronephrosis emerged as the most prevalent urological anomaly detected through voiding cystourethrogram (VCUG) and ultrasonography (US) screenings among infants. Gentamicin, as a non-carbapenem alternative, emerged as a favorable option for treating pediatric ESBL urinary tract infections (UTIs), exhibiting promising efficacy. Additionally, nitrofurantoin demonstrated efficacy as an oral antibiotic for managing uncomplicated ESBL UTIs, offering a valuable therapeutic option in such cases.

## Figures and Tables

**Table 1 antibiotics-13-00547-t001:** Population characteristics.

Variables	Total	ESBL	Non-ESBL Urinary Organisms	*p*-Value †
	(*n* = 459)	(*n* = 133)	(*n* = 326)	
	*n* (%)	*n* (%)	*n* (%)	
Gender				0.888
Male	161 (35)	46 (34.6)	115 (35.3)	
Female	298 (65)	87 (65.4)	211 (64.7)	
Ethnicity				0.257
Non-Qatari	379 (82.6)	114 (85.7)	265 (81.3)	
Qatari	80 (17.4)	19 (14.3)	61 (18.7)	
Age Group				0.184
>60 days	403 (87.8)	121 (91)	282 (86.5)	
≤60 days	56 (12.2)	12 (9)	44 (13.5)	
Gestational Age				0.208
≥37 weeks	321 (87.5)	80 (82.5)	241 (89.3)	
32–36 weeks	39 (10.6)	14 (14.4)	25 (9.3)	
≤31 weeks	7 (1.9)	3 (3.1)	4 (1.5)	
Male circumcision				0.857
Not circumcised	69 (68.3)	14 (70)	55 (67.9)	
Circumcised	32 (31.7)	6 (30)	26 (32.1)	

† Statistical significance was set at *p* < 0.05.

**Table 2 antibiotics-13-00547-t002:** Prevalence of urinary organisms.

Urine Organism	ESBL		Non-ESBL Urinary Organisms	
	*n* = 133	%	*n* = 326	%
*Escherichia coli*	106	79.7	187	57.2
*Klebsiella pneumoniae*	13	9.8	61	18.7
*Enterococcus faecalis*	-	-	28	8.6
*Proteus mirabilis*	0	0	10	3.1
*Citrobacter koseri*	0	0	9	2.8
*Pseudomonas aeruginosa*	0	0	9	2.8
*Streptococcus agalactiae*	-	-	8	2.5
*Klebsiella oxytoca*	0	0	5	1.5
*Enterobacter cloacae*	4	3	1	0.3
*Citrobacter freundii*	3	2.3	0	0
*Staphylococcus aureus*	-	-	3	0.9
*Serratia marcescens*	1	0.8	2	0.6
*Citrobacter amalonaticus*	1	0.8	0	0
*Enterobacter gergoviae*	1	0.8	0	0
*Enterococcus raffinosus*	0	0	1	0.3
*Klebsiella aerogenes*	1	0.8	0	0
*Klebsiella ozaenae*	0	0	1	0.3
*Morganella morganii*	1	0.8	1	0.3
*Proteus hauseri*	0	0	1	0.3
*Streptococcus gallolyticus*	-	-	1	0.3

**Table 3 antibiotics-13-00547-t003:** Clinical history and laboratory parameters between ESBL and non-ESBL urinary organisms.

	ESBL (*n* = 133)		Non-ESBL (*n* = 326)		*p*-Value †
Laboratory Parameters and Clinical History	Median	IQR	Median	IQR	
WBCs	13.3	8.3–18.3	12	8.6–16.8	0.251
ANCs	5.8	2.6–9.3	4.9	2.8–8.1	0.434
Platelets	407	303–476	373	301–454	0.205
CRP (mg/L)	29	6.3–55	17.4	5–52	0.446
Na level	138	136–139	137	136–138	0.314
BUN level	2.7	1.9–3.7	3	2.1–3.9	0.124
Creatinine level	20	17–23	22	19–26	<0.001
	*n*	%	*n*	%	*p*-value
Urine WBCs (>9)	105	80.8%	209	64.9%	<0.001
UA leukocytes (≥trace)	71	54.2%	135	42.5%	0.023
UA nitrites	51	38.9%	77	24.2%	0.002
NICU admission	31	29.2%	61	21.9%	0.129
Previous hospitalization	12	10.2%	26	8.4%	0.576
Previous surgeries	0	0%	2	0.6%	NA
Presence of renal/urogenital anomalies/diseases	10	9.4%	17	6.4%	0.303
Presence of urological anomalies using VCUG	3	5.7%	6	5.1%	0.572
Abnormal US result	11	12%	24	12.8%	0.847
Circumcised	6	30%	26	32.1%	0.857
Previous positive urine culture	10	8.3%	26	9.8%	0.637

Note: † *p* < 0.05. Abbreviations: WBCs—white blood cells; ANCs—absolute neutrophil count; CRP—C-reactive protein; BUN—blood urea nitrogen; VCUG—voiding cystourethrogram; US—ultrasound.

**Table 4 antibiotics-13-00547-t004:** Ultrasound urological abnormality across ESBL and non-ESBL urinary organisms.

Abnormality	ESBL	Non-ESBL Urinary Organisms
	*n* (%)	*n* (%)
Hydronephrosis	10 (10.9)	19 (10.1)
Dysplastic kidney	0 (0)	1 (0.5)
Horseshoe kidney	0 (0)	1 (0.5)
Ectopic kidney	0 (0)	2 (1.1)
Duplex collecting system	1 (1.1)	1 (0.5)

**Table 5 antibiotics-13-00547-t005:** Antimicrobial use and administration routes for ESBL and non-ESBL urinary organisms.

Organism		*n*	Route (%)		† (IV) Antibiotic (%)										(Oral)					
			Oral *n* (%)	† IV *n* (%)	Ceftriaxone	Gentamicin	Amoxicillin	Ampicillin	Tazocin ***	Cefepime	Cefotaxime	Ertapenem	Amikacin	Meropenem	Cefixime	Amox/Clav *	Cefuroxime	Nitrofurantoin	TMP–SMX **	Cefdinir
**Non-ESBL**	*Escherichia coli*	187	132 (70.6)	55 (29.4)	21	3.8	0.5		0.5		2.2				51.1	14	3.2	2.2	1.6	
	*Klebsiella pneumoniae*	61	40 (65.6)	21 (34.4)	20	3.3				1.7	5		1.7		41.7	20	5		1.7	
	*Enterococcus faecalis*	28	17 (60.7)	11 (39.3)	23.1		7.7	7.7							30.8	19.2		11.5		
	*Proteus mirabilis*	10	8 (80)	2 (20)	10										30	40	20			
	*Citrobacter koseri*	9	6 (66.6)	3 (33.3)	33.3										55.6	11.1				
	*Pseudomonas aeruginosa*	9	1 (11.1)	8 (88.9)	11.1	66.7				11.1					11.1					
	*Streptococcus agalactiae*	8	1 (12.5)	7 (87)	42.9			14.3			14.3				14.3	14.3				
	*Klebsiella oxytoca*	5	2 (40)	3 (60)	60										40					
	*Staphylococcus aureus*	3	1 (33.3)	2 (66.7)	66.7										33.3					
	*Serratia marcescens*	2	1 (50)	1 (50)		50													50	
	*Enterobacter cloacae*	1	1 (100)	0 (0)											100					
	*Enterococcus raffinosus*	1	1 (100)	0 (0)											100					
	*Klebsiella ozaenae*	1	1 (100)	0 (0)											100					
	*Morganella morganii*	1	1 (100)	0 (0)											100					
	*Proteus hauseri*	1	1 (100)	0(0)												100				
	*Streptococcus gallolyticus*	1	0 (0)	1(100)	100															
	Total	328	214 (65.6)	114 (34.4)	21.9	4.9	0.9	0.9	0.3	0.6	2.5		0.3		45.1	15.4	3.4	2.2	1.5	
**ESBL**	*Escherichia coli*	106	45 (42.5)	61 (57.5)	8.6	36.2					1	1.9	6.7	2.9	23.8	1		13.3	4.8	
	*Klebsiella pneumoniae*	13	5 (38.5)	8 (61.5)	15.4	38.5	7.7						7.7		23.1				7.7	
	*Enterobacter cloacae*	4	2 (50)	2 (50)		50												25	25	
	*Citrobacter freundii*	3	2 (66.7)	1 (33.3)										33.3	33.3				33.3	
	*Serratia marcescens*	1	0 (0)	1 (100)		100														
	*Citrobacter amalonaticus*	1	0 (0)	1 (100)	100															
	*Enterobacter gergoviae*	1	0 (0)	1 (100)		100														
	*Klebsiella aerogenes*	1	0 (0)	1 (100)		100														
	*Morganella morganii*	1	1 (100)	0 (0)																100
	Total	131	55 (43.1)	76 (56.9)	9.1	36.4	0.8				0.8	1.5	6.1	3	22.7	0.8		11.4	6.8	0.8

Note: † IV—intravenous. * Amoxicillin/clavulanate; ** trimethoprim/sulfamethoxazole; *** piperacillin/tazobactam.

## Data Availability

Upon reasonable request, the primary author can provide the raw data supporting the conclusions of this article.

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
