# Peer review of "Urinary Tract Infections among Febrile Infants in Qatar: Extended-Spectrum-Beta-Lactamase (ESBL)-Producing Versus Non-ESBL Organisms"

_antibiotics, 2024, doi:10.3390/antibiotics13060547_

Round 1
Reviewer 1 Report
Comments and Suggestions for Authors
The present study proposes, from the title, an analysis of UTI in infants in Qatar.
1. First of all the study included mostly non-Qatar patients. How is this relevant to local prevalence data?
2. Maybe it would be useful to make a comparison between local data and other populations.
3. Can you document all patients origins for a comparison to other area patterns? You might take advantage of the fact that Qatar is a high visited place.
4. You should include CLSI criteria.
5. You should compare your results with more studies in the Discussion section.
6. Please avoid personal expressions like our data, study etc...
7. In the first lines of the Material and Methods section, you gave the impression that the data were collected from more centers, but in the end, you specify in the limitation that only one center's data were analyzed. Please explain. routinely collected hospital data 67 of febrile infants who visited pediatric emergency centers in the state of Qatar over a pe-68 riod of 12 months from January to December 2019 / The data reviewed was confined to a single medical center and spanned only one 113 year, potentially limiting its generalizability to the broader national landscape of micro-114 bial etiology and antibiotic resistance in ESBL-UTI
Comments on the Quality of English LanguageMinor English issues
Author Response
Thank you for your comments
The reply to the comments has been attached.

Reviewer 2 Report
Comments and Suggestions for Authors
This work could be interesting, but it lacks important things, particularly in materials and methods and in the concepts of the works.
ESBL and non-ESBL, are Gram negative species?or all the species identified were used?I think that only Enterobacteriaceae data must be used in the analysis and graphics are necessary to show the sensitivity patterns.
The species names MUST be in Italic.
N has to be depicted as "n" in italic
Identification and antimicrobial patterns, how are they determined?
The patients are inpatients or oupatients?
The tables must be rewritten...see species names!
I think that a work about the sensitivity pattern or the prevalence of ESBL organisms can be more interesting and easy to understand!
Comments on the Quality of English LanguageModerate English revision is needed, particulary in the words used. In a research work has to be used scientific language
Author Response
Thank you for your comments
The reply comments to reviewer have been attached as word documents

Reviewer 3 Report
Comments and Suggestions for Authors
24-05-19-Qusad 2024 subm ab-Review-UTI among Febrile Infants in Qatar: ESBL producing bacteria Vs Other Organisms
This is an interesting retrospective cohort study during one year on febrile UTIs among 459 infants. E. coli and K. pneumoniae were the most frequent pathogens in the ESBL and in the non-ESBL group. The authors compared both groups concerning demographics, pathogens and treatment options. The study is well performed and presented.
I only have a few minor comments/questions:
1. ESBL – extended spectrum betalactamase. Somtimes the authors use as abbreviation EBSL, which of course is wrong.
2. The authors differentiate the pathogens/microorganisms between ESBL group and non-ESBL group. This is of course correct. The differentiation between ESBL group and other organisms is misleading. Other organisms could mean , which do not produce or which cannot produce betalactamases, such as E. faecalis, S. agalactiae, etc. Therefore I recommend, that the authors always should use only the terms: ESBL group and non-ESBL group to avoid confusion.
Page 5, last third line: „complicated ESBL group“ – do you mean ESBL group with complicated UTI (not the ESBL is complicated, but UTI is complicated)
Page 6, line 4: „In contrast, for the non-ESBL and other organisms‘ group, Cefixime…“ What do you mean? Is this the same group or two different groups, which is not clear. Therefore, always it should be the terms: ESBL group and non-ESBL group also in the Tables.
Title: Urinary Tract Infections among Febrile Infants in Qatar: Extended-Spectrum Beta-Lactamase (ESBL) producing versus non-ESBL Organisms.
I have no further comments.
Comments on the Quality of English Languagesee comments for authors
Author Response
Thank you for your comments
The reply to the comment has been attached

Round 2
Reviewer 1 Report
Comments and Suggestions for Authors
No further comments
Reviewer 2 Report
Comments and Suggestions for Authors
The species names MUST be written in Italic!
correct the manuscript and the tables!